# The Quality in Psychiatric Care–Inpatient Staff Instrument: A Psychometric Evaluation

**DOI:** 10.3390/healthcare10071213

**Published:** 2022-06-28

**Authors:** Agneta Schröder, Lars-Olov Lundqvist

**Affiliations:** 1University Health Care Research Center, Faculty of Medicine and Health, Örebro University, 701 82 Örebro, Sweden; lars-olov.lundqvist@regionorebrolan.se; 2Department of Health Science, Faculty of Health, Care and Nursing, Norwegian University of Science and Technology (NTNU), 2815 Gjövik, Norway

**Keywords:** inpatient care, instrument, psychiatry, psychometric properties, staff, quality of care

## Abstract

Much work has focused on the development of instruments that measure the quality of care, but few studies have been published for staff assessment of the quality of care provided by inpatient psychiatric care. Therefore, an instrument is needed to measure the quality of care from the perspective of facility staff. The aim of the present study was to evaluate the psychometric properties and factor structure of the Quality in Psychiatric Care-Inpatient Staff (QPC-IPS) instrument. A sample of 104 staff at seven wards in four regions in Sweden completed the QPC-IPS, which consists of 30 items covering six dimensions of quality. Confirmatory factor analysis confirmed the proposed six factor structure of the QPC-IPS. Internal consistency for the full QPC-IPS was adequate, but poor for some of the dimensions. Staff ratings of the quality of care were generally high. The highest rating was for the *Support* dimension and the lowest for the *Secure* environment dimension.

## 1. Introduction

Quality of care is a topic relevant in health management, research, and clinical practice. In psychiatric care, the interest in issues of quality-of-care measurements has been growing [1]. The use of well-constructed measurement instruments with high levels of reliability and validity is an effective way to measure the quality of care [2], and their use is fundamental for improving the quality of psychiatric care and finding out where changes are needed [3].

Patients’ experiences of quality of care have for many years been deemed to be an important measure of quality of care [3]. The assessment of the staff’s judgement of the quality of care can also provide useful data for improving and developing the care and its efficacy. This data is important as it reflects the views of the staff providing the care, which is a significant indicator when measuring its quality [4].

Extensive work has been focused on developing quality of care measuring instruments in the health care services [5]. In spite of this, less has been published on how staff experience the quality of psychiatric inpatient care. In a systematic review of the psychometric properties of instruments measuring quality of care in psychiatric care, Sanchez-Balcells et al. [6] found only five measurements for staff/management, and of these, only three were for inpatient care, namely for management [7], for nursing staff and patients [8] and for forensic inpatient care staff [9]. Only one of these instruments measured how staff experience the quality of inpatient care, but this instrument was for forensic inpatient care staff. In order to systematically assess the quality of care from the perspective of the staff, valid standardised measurements are needed that can aid comparative benchmarking both nationally and internationally. Such measurements can also identify incongruence between the views of patients and staff on the quality of care [10], as evidence shows that understandings and perspectives on efforts can vary between the two groups [11]. Hence, both perspectives are needed to obtain a more complete assessment [6].

Instruments measuring quality of care should have a clear conceptualisation of what quality of care is [12]. As quality of care is context specific [13], measuring instruments should also be designed for the services where they are to be used [14]. The Quality in Psychiatric Care-Inpatient (QPC-IP) instrument was developed from a definition of quality of care, psychometrically tested, context specific and is frequently used for measuring quality of care from the patients’ perspective [15]. Recently, the Quality in Psychiatric Care-Inpatient Staff (QPC-IPS) instrument was developed for measuring staff perceptions of the quality of care with the same items as the QPC-IP, providing the possibility of comparing the two perspectives on the ratings for quality of care. Still, little is known about how staff and patients sharing the same ward perceive the quality of care. One reason can be the lack of a single instrument that covers the same dimensions of quality of care for both the patients and staff [16]. When a new instrument is developed, it is necessary to develop, refine and establish its psychometric properties [17]. Therefore, the aim of the present study is to evaluate the psychometric properties and factor structure of the Quality in Psychiatric Care-Inpatient Staff (QPC-IPS) instrument.

## 2. Materials and Methods

### 2.1. Design

The study has a cross-sectional design.

### 2.2. Participants and Procedures

A sample of 104 staff at seven wards in four regions in Sweden completed the QPC-IPS. Data collection took place during six months in 2019. Inclusion criteria for participants encompassed: (1) permanent employees or locum tenens in psychiatric inpatient care and (2) having worked for at least three months in psychiatric inpatient care. The invitation to participate was handled by a contact person (mostly the head of the ward) who orally and in writing informed the staff about the aim of the study. All participants were informed that their participation was voluntary, that their answers would be anonymised and that they could withdraw at any time. Those who agreed to participate gave their oral consent and were requested to complete the QPC-IPS instrument anonymously.

### 2.3. The QPC-IPS Instrument

The QPC, short for Quality in Psychiatric Care, is a self-reported and multidimensional instrument developed for patients’ perceptions of quality of care [16] and adapted to six patient versions for diverse mental health contexts. Each patient version has a common core with context-specific items [18] and a staff version with the same dimensions and items. Currently, the instrument for the inpatient care staff (QPC-IPS) in Indonesian and Spanish [19,20] and forensic inpatient care staff in Swedish and Danish (QPC-FIPS) [9,21] have been psychometrically evaluated.

In this study, staff perceptions were measured with the Swedish QPC-IPS, short for Quality in Psychiatric Care-Inpatient staff. The QPC-IPS starts with a number of background questions concerning demography and general relevant information, followed by 30 items divided into six quality dimensions: Encounter (8 items), Participation (8 items), Discharge (4 items), Support (4 items), Secluded environment (3 items), and Secure environment (3 items). Each item describes perceptions of the provided quality of care and consists of a statement on quality of care (e.g., "I experience that the patients are involved in decisions about their care") rated on a 4-point Likert type scale, ranging from 1 = totally disagree to 4 = totally agree with an additional non-applicable response alternative. An open question at the end invites further comments on the quality of care.

### 2.4. Data Analysis

To compute descriptive statistics, we used the statistical software package SPSS version 22 (IBM Corp., Armonk, NY, USA). To assess the scales, we used Cronbach’s alpha [22] with Nunnally and Bernstein’s [23] 0.70 criterion for adequate homogeneity. Confirmatory factor analysis (CFA) was performed using LISREL 8.8, Scientific Software International: Chicago, IL, USA [24] with generally weighted least-squares estimation on the asymptotic covariance matrices. The PRELIS program [25] obtained polychoric and polyserial correlation matrices and the weighted least squares method using the asymptotic covariance matrix estimated the parameters. The Satorra–Bentler scaled chi-square (S-Bχ^2^) test [26] was used because the data were ordinal (i.e., 4-point Likert scale).

Confirmatory factor analysis was used to test whether the factor structure, i.e., the construct validity of the QPC-IPS is consistent with that of the OPQ-IP, serving as an a priori model. Thus, items 7, 10, 11, 12, 15, 18, 20 and 25 were assumed to represent Encounter; items 1, 5, 6, 13, 14, 27, 29 and 30 represent Participation; items 8, 16, 17 and 21 represent Discharge; items 19, 22, 23 and 24 represent Support; items 3, 26, and 28 represent Secluded environment; and items 2, 4 and 9 represent Secure environment.

In addition to the Satorra–Bentler scaled chi-square (S-Bχ^2^), we used the root mean square error of approximation (RMSEA), the standardised root mean square residual (SRMR), and the comparative fit index (CFI) to evaluate the adequacy of the a priori model [27]. Values lower than 0.08 and 0.05 for the RMSEA, equal to or lower than 0.10 and 0.08 for the SRMR, and equal to or greater than 0.90 and 0.95 for the CFI were considered to constitute an adequate and excellent level of goodness of fit, respectively [28].

## 3. Results

### 3.1. Sample Description

The sample consisted of 104 staff (81 women and 23 men) aged between 22 and 67 years (mean = 46.8, SD = 14.2), of which 71 (68%) were nursing assistants, 28 (27%) were licensed nurses, and the remaining 5 (5%) were psychiatrists, counsellors or care assistants. Of the staff, 100 (96%) were Swedish and the remaining 4 (4%) were from other European countries. The staff had worked at the ward for on average 7.7 years (range 0–43 years) and most of them—68 (65%)—worked daytime, 20 (19%) worked a combination of day- and night-time, and 14 (14%) worked night-time only.

### 3.2. Factor Structure of the QPC-IPS

The CFA performed on the model representing the QPC-IP factor structure showed a significant chi square (S-Bχ^2^ = 517.67, df = 390, *p* < 0.001), CFI = 0.98, RMSEA = 0.056 (CI = 0.043–0.069) with a Close Fit (RMSEA < 0.05) *p* = 0.21, and SRMR = 0.092. This result indicated an adequate to excellent fit. Hence, the model was regarded an acceptable representation of the proposed factor structure of the QPC-IPS. All loadings were significant (*p* < 0.05).

### 3.3. Internal Consistency

The internal consistency of the QPC-IPS was adequate with a Cronbach α of 0.94 for the full questionnaire as well as for the individual factors Encounter (α = 0.90), Participation (α = 0.87), Discharge (α = 0.82), Secure environment (α = 0.76), and Support (α =0.75). Less than adequate consistency was, however, found for Secluded environment (α = 0.65).

### 3.4. Test Retest

Forty staff completed the QPC-IPS twice and test-retest analyses demonstrated excellent retest reliability for the full QPC-IPS, and adequate to excellent retest reliability for the specific dimensions, with Participation receiving the highest and Secluded environment the lowest retest reliability (Table 1).

### 3.5. Description of Staff View of Inpatient Quality of Care

The mean and standard deviations for the QPC-IPS items are given in Table 2.

As shown in Figure 1, and supported by dependent t-tests, the perceived quality of Support was significantly greater than the perceived quality of the second-ranked dimension, Encounter (t(103) = 3.46, *p* < 0.001), which in turn was perceived as significantly greater than Discharge (t(103) = 9.90, *p* < 0.001). Discharge was perceived as greater than Participation (t(103) = 3.49, *p* < 0.001), which, however, was not perceived as significantly greater than Secluded environment, which in turn was significantly larger than Secure environment (t(103) = 2.14, *p* = 0.035).

As shown in Table 3, all correlations among the QPC-IPS dimensions were significantly correlated with coefficients demonstrating moderate to strong relationships.

## 4. Discussion

Measuring quality of care is a step towards improving psychiatric inpatient care but it has been held back by a lack of psychometrically tested instruments for measuring staff perception of the quality of care provided. Therefore, the aim of the present study was to evaluate the psychometric properties and factor structure of the QPC-IPS. The findings demonstrate that the QPC-IPS is a valid and reliable instrument for measuring staff perceptions of the quality of care in psychiatric inpatient care. The CFA showed that the factor structure of the QPC-IPS was equivalent with that of the patients’ version (QPC-IP) [15], demonstrating its construct validity. Thus, both patients and staff in inpatient psychiatric care have a similar view of the quality of care concept expressed in the QPC instruments.

Even though the aim of the study was to evaluate the psychometric properties and factor structure of the QPC-IPS, it is worth noting that the staff in this study perceived the quality of care in general as high: 88.4% had a mean score greater than 2.5, which is the centre of the rating scale used in QPC-IPS. This is in line with other studies using the QPC instruments in various psychiatric cultures and contexts, for example, Denmark [21], Indonesia [19] and Spain [20]. The highest rating in the present study was in the dimension of Support, which is in line with previous studies on staff in inpatient care [19,20]. The findings from the current study may indicate that staff were aware of the patients’ shame and guilt and sought to prevent these feelings in their treatment. This finding also indicates that the staff prevents situations where the patient becomes a threat to themselves and others. This is strange and may be contradictory, as the lowest rating was in the Secure environment dimension. This finding highlights that staff did not experience the care environment as safe and that the patients did not feel secure among their fellow patients. This is in line with previous studies on perceptions of quality among staff in forensic inpatient care in both Sweden [9] and Denmark [21] as well as in other studies regarding safety in psychiatric inpatient wards [29,30]. It seems that staff in inpatient care, regardless of context, perceived the ward environment as unsafe. Maybe this can be explained by the occurrence of violence, mainly verbal and sometimes also physical, in psychiatric care [31]. Regardless, staff should create a secure foundation for patients in inpatient care [32] and the safety of both patients and staff is an important aspect of the quality of care [31]. In addition, nurses are expected to apply research evidence in practice to improve both the quality of care and safety on the ward [33], as they are key personnel in the delivery of health care [34]. Thibaut et al. [35] conclude in a systematic review that patient safety in inpatient psychiatric care is under-researched, therefore more research is needed in this area.

### 4.1. Limitations

Some limitations of the study must be mentioned. Only 104 staff agreed to participate in this study. The general rule of thumb [36] suggests a minimum of 150 respondents when the instrument consists of 30 items. On the other hand, finding participants for questionnaire-based studies is a general problem [37]. In this study, it was not possible to analyse the dropouts because of the incomplete registration of “missing staff” at the wards.

### 4.2. Implications for Practice

The QPC-IPS is an adequate instrument for measuring quality of care from the staff’s perspective and can be used as a tool in daily work. This is important information as staff, especially nurses, are crucial in the process of implementing measurements.

Generally, the scores for the different dimensions indicated where improvements may be required, and the QPC-IPS can be used as a local, regional and national self-rating instrument. The awareness of the staff’s low ratings for the quality of care regarding the dimension Secure environment can be used to manage and support quality of care improvement.

## 5. Conclusions

The 30-item QPC-IPS is a psychometrically adequate instrument and is an easy, time-efficient, inexpensive and quick way to evaluate quality of care. QPC-IPS can be used as a tool to motivate care staff to continually and systematically improve the care provided. The low rating in the dimension Secure environment is a challenge for nursing staff as well as other staff and management to improve. In addition, the instrument can with advantage be used with the patient version (QPC-IP) [15] to achieve an overall picture of the quality of care. As Shannon, Mitchell, and Cain [38] state, both perspectives are needed. Therefore, further research should compare both the patients’ and the staff’s ratings of quality of care. So far, QPC-IPS is available, apart from Swedish, in two languages—Indonesian [19] and Spanish [20]—which allows for international comparisons of quality of care.

## Figures and Tables

**Figure 1 healthcare-10-01213-f001:**
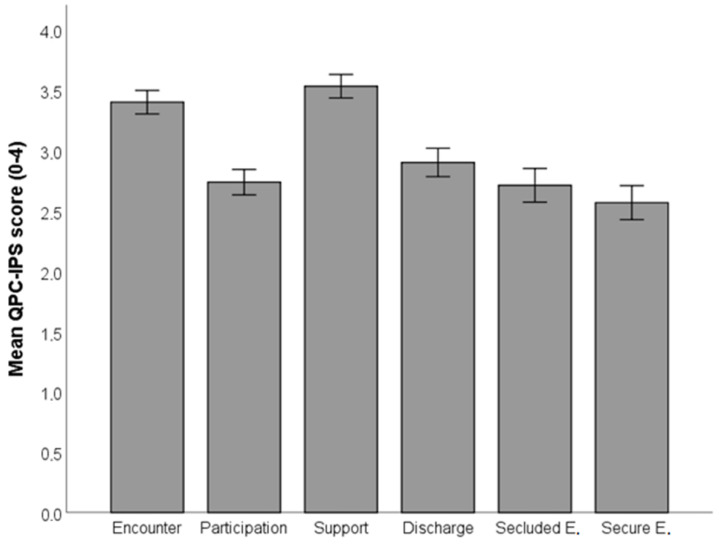
Mean QPC-IPS scores for each QPC-IPS dimension.

**Table 1 healthcare-10-01213-t001:** Test-retest statistics for the QPC-IPS among staff in inpatient psychiatric care at Time 1 (T1) and Time 2 (T2) × months later.

	QPC-IPS Dimension	M_T1_	SD_T1_	M_T2_	SD_T2_	ICC	95% CI
1.	Encounter	3.39	0.43	3.45	0.38	0.89	0.84–0.94
2.	Participation	2.65	0.56	2.64	0.48	0.92	0.87–0.95
3.	Discharge	2.82	0.60	2.80	0.57	0.82	0.72–0.89
4.	Support	3.53	0.50	3.53	0.46	0.89	0.84–0.94
5.	Secluded Environment	2.63	0.70	2.67	0.60	0.75	0.61–0.86
6.	Secure Environment	2.58	0.63	2.64	0.61	0.84	0.75–0.91
	Total QPC	2.98	0.42	3.00	0.36	0.95	0.93–0.97

N = 40. ICC, intraclass correlation. CI, confidence interval.

**Table 2 healthcare-10-01213-t002:** Summary statistics for the QPC-IPS.

	QPC-IPS Items		Loadings	Alpha	Mean	SD
		**Total QPC-IPS (30 items)**	0.94		3.03	0.46
	**1. Encounter (8 items)**			0.90	3.40	0.50
7.	Patients have the opportunity to talk when needed		0.67		3.29	0.72
10.	Staff are involved		0.83		3.44	0.68
11.	Staff treat the patients with warmth and consideration		0.88		3.48	0.64
12.	Staff care if the patients get angry		0.88		3.37	0.71
15.	Staff respect the patients		0.89		3.58	0.57
18.	Staff show they understand the patients’ feelings		0.81		3.28	0.66
20.	Staff have time to listen to the patients		0.69		3.14	0.73
25.	Staff are concerned about the patients’ care		0.86		3.64	0.54
	**2. Participation (8 items)**			0.87	2.74	0.55
1.	Patients can influence their own care and treatment		0.75		2.60	0.70
5.	Patients’ opinion of the right care is respected		0.76		2.70	0.71
6.	Patients are involved in decisions about their care		0.75		2.67	0.74
13.	Benefit drawn from earlier experience of treatment		0.64		2.80	0.76
14.	Patients get to recognise signs of deterioration		0.75		2.65	0.76
27.	Patients receive information in a way that they can understand		0.80		2.96	0.72
29.	Patients are informed about their mental health problems		0.79		2.83	0.74
30.	Patients receive information about treatment alternatives		0.67		2.71	0.88
	**3. Discharge (4 items)**			0.82	2.90	0.61
8.	Patients are offered planning of their continued treatment		0.81		2.94	0.80
16.	Patients are offered a follow-up after discharge		0.66		3.15	0.79
17.	Patients are given help in finding an occupation		0.76		2.31	0.88
21.	Patients know where to turn after discharge		0.67		3.21	0.73
	**4. Support (4 items)**			0.75	3.54	0.50
19.	Staff prevent the patients from hurting each other		0.66		3.52	0.61
22.	Staff prevent the patients from hurting themselves		0.73		3.47	0.61
23.	Nothing shameful about having mental problems		0.98		3.64	0.61
24.	Staff tell patients shame must not interfere with seeking treatment		0.87		3.51	0.67
	**5. Secluded environment (3 items)**			0.65	2.71	0.72
3.	Patients have access to a place that is private		0.71		2.83	1.05
26.	Patients have their own room		0.66		2.25	0.79
28.	Private place where patients can receive visits from family		0.68		3.07	0.96
	**6. Secure environment (3 items)**			0.76	2.57	0.73
2.	Security is high at the ward		0.66		2.75	1.00
4.	Patients feel secure with fellow patients		0.93		2.73	0.82
9.	Patients are not disturbed by fellow patients		0.79		2.23	0.82

N = 104. All loadings are statistically significant (*p* < 0.05).

**Table 3 healthcare-10-01213-t003:** Correlation coefficients for the QPC-IPS dimensions among staff in inpatient psychiatric care.

	QPC-IPS Dimension	1	2	3	4	5	6
1.	Encounter	1.00					
2.	Participation	0.67	1.00				
3.	Discharge	0.58	0.78	1.00			
4.	Support	0.70	0.56	0.44	1.00		
5.	Secluded environment	0.46	0.67	0.57	0.31	1.00	
6.	Secure environment	0.54	0.59	0.48	0.34	0.55	1.00

N = 104. All correlation coefficients are significant with a *p* < 0.01.

## Data Availability

Data available on request from authors. The data that support the findings of this study are available from the corresponding author upon reasonable request and in accordance with ethical approval.

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
