# Peer review of "The Quality in Psychiatric Care–Inpatient Staff Instrument: A Psychometric Evaluation"

_healthcare, 2022, doi:10.3390/healthcare10071213_

Round 1
Reviewer 1 Report
Congratulations
Additional comments:
1. What is the main question addressed by the research?
2. Do you consider the topic original or relevant in the field, and if
so, why?
1-2. The research is aimed at answering a question aimed at examining and predicting an instrument that evaluates the feedback on the quality of care by the staff of a psychiatric inpatient clinic.
The aspect of the study, which aims to evaluate the quality of care from the perspective of the caregiver, will test the need for an instrument that can provide insight to the caregiver, and in this respect, it will provide clues that will contribute to increasing the quality of care. This is important.
3. What does it add to the subject area compared with other published
material?
3. As far as I know, factor analysis and sub-dimensions of this instrument have not been studied in detail. In this respect, I believe that it will make an additional contribution to the literature.
4. What specific improvements could the authors consider regarding the
methodology?
4. I don't have any change proposals or suggestions about methodology.
5. Are the conclusions consistent with the evidence and arguments
presented and do they address the main question posed?
5. I think the results are consistent enough.
6. Are the references appropriate?
7. Please include any additional comments on the tables and figures.
6-7. The references, tables and figures are appropriate.
Author Response
Thank you very much for your through review.
Reviewer 3 Report
Good day and thank you for the oppurtunity to review the manuscript "The Quality in Psychiatric Care – Inpatient Staff instrument: A psychometric evaluation". This psychometric evaluation of QPC-IPS is clearly presented albeit the size of the sample is small as mentioned in the limitations.
I have minor point:
1. The manuscript focuses on the reliability of the instrument but perhaps the valitidity could be mentioned in the discussion. Despite the problem of gathering self-reported data on the care given by the personal themselves, a reliable intstrument of measuring their attitude is clearly of importance but hard to understand if the quality of care is good or only that healthcare professionals see it that way.
2. On the same theme as comment 1 - would you consider it a strength or a weakness to have the distribution of professions as presented since 95% were either nursing assistants or nurses?
3. There is a small typo in table 2 where the score for the overall scale is in the wrong column.
Author Response
Than you for your thorough review
Please see the attachment

Reviewer 4 Report
Line 35. The authors speak about the extensive work that has been done (...) and only refer to one study.
Line 68. Was the contact person who collected the sample responsible for recruiting the 104 participants? Did he/she know them personally? Biases in the study should be discussed.
How many were initially proposed to be part of the study and how many finally participated?
Author Response
Thank you very much for your thorough review.
Please see the attachment,

Round 2
Reviewer 4 Report
Dear authors, thank you for the changes you have made, which significantly improve the text.
This manuscript is a resubmission of an earlier submission. The following is a list of the peer review reports and author responses from that submission.